# A Sensitivity-Optimized Flexible Capacitive Pressure Sensor with Cylindrical Ladder Microstructural Dielectric Layers

**DOI:** 10.3390/s23094323

**Published:** 2023-04-27

**Authors:** Tian Hua, Ziyin Xiang, Xiangling Xia, Zhangling Li, Dandan Sun, Yuanzhao Wu, Yiwei Liu, Jie Shang, Jun Chen, Runwei Li

**Affiliations:** 1School of Materials Science and Engineering, Jiangxi Provincial Key Laboratory of Power Batteries and Materials, Jiangxi University of Sciences and Technology, Ganzhou 341000, China; 2CAS Key Laboratory of Magnetic Materials and Devices, Ningbo Institute of Materials Technology and Engineering Chinese Academy of Sciences, Ningbo 315201, China; 3Zhejiang Province Key Laboratory of Magnetic Materials and Application Technology, Ningbo Institute of Materials Technology and Engineering Chinese Academy of Sciences, Ningbo 315201, China; 4College of Materials Science and Opto-Electronic Technology, University of Chinese Academy of Sciences, Beijing 100049, China

**Keywords:** flexible capacitive pressure sensor, dielectric layers microstructural, sensitivity-optimized

## Abstract

Flexible capacitive pressure sensors have attracted extensive attention due to their dynamic response and good sensing capability for static and small pressures. Using microstructural dielectric layers is an effective method for improving performance. However, the current state of microstructure design is primarily focused on basic shapes and is largely limited by simulation results; there is still a great deal of potential for further innovation and improvement. This paper innovatively proposes to increase the ladder structure based on the basic microstructures, for example, the long micro-ridge ladder, the cuboid ladder, and cylindrical ladder microstructures. By comparing 9 kinds of microstructures including ladder structure through finite element simulation, it is found that the sensor with a cylindrical ladder microstructure dielectric layer has the highest sensitivity. The dielectric layers with various microstructures are obtained by 3D printed molds, and the sensor with cylindrical ladder microstructure dielectric layer has the sensitivity of 0.12 kPa^−1^, which is about 3.9 times higher than that without microstructure. The flexible pressure sensor developed by us boasts sensitivity-optimized and operational stability, making it an ideal solution for monitoring rainfall frequency in real time.

## 1. Introduction

The skin is the largest organ of the human body and serves as a bridge between the body and the external environment. It allows humans to perceive external temperature and pressure, as well as the surface shape and texture of objects. Additionally, it performs functions such as protection, excretion, regulation of body temperature, and sensing external stimuli. Building on the capabilities of the skin, researchers have been working to develop electronic skins that mimic human skin [1,2]. While human skin transmits external information to the brain through nerve fibers, electronic skin utilizes flexible sensor technology to convert detected changes into electrical signals and feedback to a recognition device. As an artificial intelligent skin, it is composed of multiple sensors distributed or stacked along the same surface, mimicking some of the characteristics of human skin and capable of endowing robots or prosthetics with tactile sensation. Furthermore, electronic skin can serve as a “second skin” for humans by adhering to the body surface, where sensors measure various physical parameters (such as blood pressure, body temperature, and heart rate) or environmental parameters (such as gases, chemicals, materials, and radiation) to enhance natural sensory abilities [3,4,5,6,7,8,9].

The ability to perceive pressure is the most basic function of electronic skin, and enhancing pressure-sensing capabilities has always been an important topic in electronic skin research [7,10,11]. In recent years, with the rapid growth of intelligent robots and wearable electronic devices, various new devices based on flexible pressure sensors are being developed at a high speed, and are being used in fields such as entertainment, gaming, automotive, consumer electronics, industrial, and healthcare. The expansion of these application areas has placed higher demands on sensor performance [12,13]. One of the most important parameters for sensors is sensitivity. For example, when detecting weak signals such as heartbeats and pulses, sensors need to have high sensitivity to accurately perceive them. Therefore, improving sensor sensitivity is essential. To enhance the sensitivity of pressure sensors, researchers have been exploring various methods, such as using novel materials, optimizing the sensor’s structure and dimensions, and developing advanced signal processing algorithms [14,15,16]. Additionally, advancements in microfabrication technology have enabled the production of high-performance sensors with small size, low power consumption, and high sensitivity.

There are various types of flexible pressure sensors, including capacitive, resistive, and piezoelectric sensors, etc. [17,18,19,20]. Among them, capacitive flexible pressure sensors have gained widespread attention due to their advantages such as good temperature stability, dynamic response, and good sensing capability for static and small pressures. The basic principle of capacitive flexible pressure sensors is to use the compression deformation of the dielectric to change the capacitance, thereby realizing the measurement of pressure. Compared with other types of flexible pressure sensors, capacitive sensors have higher sensitivity and stability, and can respond quickly and accurately to pressure changes [14,21,22,23,24,25].

Currently, the most commonly used methods to enhance the sensitivity of capacitive pressure sensors include increasing the dielectric constant of the dielectric layer, and modifying the structure of the dielectric layer [25,26,27]. However, adding high dielectric constant substances may compromise the flexibility of the sensor, while highly conductive materials like silver and carbon nanotubes are expensive and challenging to prepare. Moreover, other metal conductive materials are prone to oxidation and damage, making the sensor more rigid. As a result, researchers have explored altering the internal structure of the sensor to decrease its elastic modulus and enable a large deformation between the plates under low pressure. This unique structure that can enhance the sensor’s sensitivity is commonly referred to as a microstructure [28,29].

Numerous studies have shown that microstructural of flexible electrode layers or dielectric layers is an effective method for improving performance. Compared to the non-structure elastomer, the air gap created by microstructures provided an improvement of hysteresis. By introducing microstructures into the dielectric or electrode layers, the compressibility and effective dielectric constant change can be improved, resulting in a larger change in capacitance for capacitive pressure sensors. Additionally, studies show that dielectric layers with microstructures are higher up the sensitivity of flexible pressure sensors. Common surface microstructures mainly include the pyramid, dome, triangular column, circular column, and square column. Voids allow the layer to easily deform upon application of pressure, the higher compressibility of the dielectric layer results in a greater change in capacitance for a given applied pressure. Therefore, sensitivity enhancement of sensors with these microstructures is evident. Guan-Jun Zhu et al. prepared a resistive pressure sensor with a hemispherical surface microstructure, with a sensitivity of 6.258 kPa^−1^ and a detection range of 0–204.7 kPa, achieving high sensitivity and wide range to a certain extent [30]. Jun Chang Yang et al. prepared a porous pyramid structure; the sensitivity of the prepared capacitive sensor was 44.5 kPa^−1^ in the detection range of 0.14 Pa–0.1 kPa, and the sensitivity of the prepared resistive sensor was 449 kPa^−1^ [31].

Researchers have conducted extensive work to understand the effects of structural design and geometric changes on sensitivity; due to limitations in fabrication processes, the systematic study of the impact of various microstructure sizes and shapes on sensitivity remains primarily in the realm of simulation [32,33,34,35]. Fortunately, 3D printing technology offers excellent design flexibility, good repeatability, and a simple fabrication process, making it possible to fabricate and compare microstructures of various shapes. With 3D printing, researchers can fabricate complex geometries that were previously impossible to produce with traditional manufacturing techniques, such as micro-lattices, metamaterials, and other types of microstructures. By using 3D printing technology, researchers can systematically investigate the effects of microstructure size, shape, and arrangement on sensitivity. 3D printing technology has the potential to revolutionize the study of microstructures and their impact on sensitivity.

In this work, we performed a sensitivity simulation comparison of sensors with 9 common microstructure shapes by COMSOL software. We found that the sensor with cylindrical ladder structure has the highest sensitivity. We also performed a comparison of the shape parameters of the cylindrical ladder structures, qualitatively predicting and comparing the effects of first-layer diameter, number of layers, layer thickness, width of each step, and separation distance on sensitivity. In order to verify the results of simulation calculations, we used 3D printing technology to prepare molds with 9 different microstructures and prepared the dielectric layer of the capacitive flexible pressure sensor by the casting method. Finally, we fabricated 9 prototypes of capacitive flexible pressure sensors with different microstructures and tested them. The experimental results further verified the simulation calculation results.

## 2. Selection of Microstructure

The flexible capacitive pressure sensor consists of upper and lower electrodes and a dielectric layer sandwiched between the electrodes. This paper presents a systematic study of dielectric layers with nine different microstructures, respectively, which include circular column, cones, pyramids, domes, cuboids, long micro-ridges, cuboid ladder, long micro-ridge steps, and cylindrical ladder, as shown in Figure 1. The base length and height of these nine microstructures are the same. Here, we take the cylindrical ladder microstructure as a representative; please refer to Appendix A, in which the length and width of the 9 microstructures are all the way 610 μm, the height is all the way 250 μm. By adding laddershaped microstructure, we explore the mechanism by which the intrinsic size of microstructures affects the sensitivity of dielectric layers. We focus on the design and preparation of ladder microstructures, which is a unique and innovative aspect compared to existing work by other researchers.

Nine microstructures were established in COMSOL, and a control model without microstructure (flat panel) was built to study the stress and stress distribution of each microstructure under a micro-displacement of 30 μm, as shown in Figure 2a. The stress distribution of each microstructure was compared with the flat panel. The closer to the warm color on the color scale, the greater the stress, modulus, and hardness of the material, and the harder it is to deform. The stress and stress distribution of the control model on the first place of Figure 2a show that its stress is higher than that of the nine microstructures. Among the microstructures, it is obvious that the square column and circular column with integrated upper and lower dimensions are more difficult to deform. Microstructures with a small size at one end but smooth transition in size, such as the long micro-ridge, dome, pyramid, and cone, have relatively small stress and smoothly distributed stress from top to bottom, and there is no stress concentration phenomenon. However, for microstructures with ladder structures, such as the long micro-ridge ladder, cuboid ladder, and cylindrical ladder, the stress is very concentrated on the top, i.e., the end with a small size, and the stress between each step is not smoothly transitioned. The stress appears to have an obvious stratification according to the ladder structure. Overall, these simulation results can provide useful information for the design and optimization of microstructures to improve the mechanical properties and stability of microstructures.

Based on simulation results, capacitance changes are calculated for 9 different microstructures and dielectric layers under stress changes ranging from 0 to 1.0 kPa. The results in Figure 2b show that the dielectric layer without microstructure (flat panel) has the lowest capacitance change rate under stress change, while the dielectric layer with microstructure cylindrical ladder has the highest capacitance change rate under stress change. An interesting phenomenon has been observed when comparing the microstructure with and without ladder structure (long micro-ridge ladder with long micro-ridge, cuboid ladder with pyramid, cone with cylindrical ladder). It was found that the ladder structure could improve the capacitance change rate under stress change for the same shape of microstructure. These results suggest that the choice of microstructure can significantly impact the performance of a dielectric layer under stress changes, with ladder structures being particularly effective in improving capacitance change rates.

Due to the computational complexity of simulating an array of microstructures, we only calculated the sensitivity of a single microstructure in the simulation. Through simulation, we got the conclusion similar with the capacitance change above, that the structural sensitivity of the flat panel without microstructure is the lowest, while the microstructure of cylindrical ladder shape has the highest structural sensitivity, as shown in Figure 2c.

In order to verify the reliability of the simulation and study the influence of mold surface microstructures on flexible capacitive pressure sensors, we fabricated molds with the above nine microstructures by 3D printing technology. Subsequently, we produced the corresponding dielectric layer structures with microstructures by casting molds. This allowed us to prepare a prototype of a flexible capacitive pressure sensor. The preparation process is shown in Figure 3.

3D printing technology has been used to create molds (1.25 × 1.25 cm^2^) with specific surface microstructures. A high-temperature resistant resin (HTL Yellow-20, BMF Precision Tech Inc., Chongqing, China) has been chosen for printing, which offers excellent stability and mechanical properties. We began by printing a mold with the desired surface microstructure by using a micro-nano processing digital light-curing 3D printer (MicroArch ^®^S240, BMF Precision Tech Inc.). Next, we placed the mold in a beaker filled with isopropanol and subjected it to ultrasonic treatment (SB-1000DTS, Ningbo Scientz Biotechnology Co., LTD., Ningbo, China) under 25,000 Hz for 5 min to ensure that there was no printing resin residue left in the microstructure imprinting groove. We allowed the mold to dry naturally and waited for the isopropanol to evaporate. Sylgard 184 elastomer-to-cross-linker was mixed in the ratio of 10:1, then the poly(dimethylsiloxane) (PDMS) mixture was poured into the mold and placed on a horizontal platform for 3 days to allow it to solidify. After that, we transferred the sample and the mold together to a blast oven at 60 °C and heated it for 7 days. This process helped to strengthen the PDMS material and increase its durability. Then, the PDMS intermediate dielectric layer with the surface microstructure had been peeled off and was attached with conductive silicone rubber electrodes (RG-067, Raycus (Dongguan) Industrial Co., Ltd., Dongguan, China) on both its upper and lower sides. Finally, we utilized PU film (Medical adhesive tape, Shanghai Hons Medical Technology Co., Ltd., Shanghai, China) for top and bottom packaging to complete the capacitive sensor.

The sensitivity obtained from the sensor prototype prepared by the above method is basically consistent with the simulation result, as shown in Appendix A Appendix A. Please note that when referring to Appendix A, because the sensitivity obtained in the experiment is measured from the sensor prototype, and for the simulation we simulate the sensitivity of a single microstructure, there are some differences between the experimental data and the simulated data, but the trend of change is consistent. The sensitivity of the cylindrical ladder structure capacitive sensor obtained by the test is 0.12136 KPa^−1^, which is about 3.9 times that of the microstructure-free sensor.

For capacitive sensors, the formula for calculating the sensitivity of the sensor is:$$S=\frac{\Delta C/C_{0}}{\Delta P}=\frac{C_{1}-C_{0}}{C_{0}\Delta P}=\frac{\frac{A\varepsilon_{r}{}_{2}\varepsilon_{0}}{d_{1}}-\frac{A\varepsilon_{r1}\varepsilon_{0}}{d_{0}}}{\frac{A\varepsilon_{r1}\varepsilon_{0}}{d_{0}}\Delta P}=\frac{\frac{\frac{\varepsilon_{r}{}_{2}}{\varepsilon_{r}{}_{1}}}{d_{1}}-\frac{1}{d_{0}}}{\frac{\Delta P}{d_{0}}}=\frac{\frac{d_{0}\frac{\varepsilon_{r}{}_{2}}{\varepsilon_{r}{}_{1}}}{d_{1}}-1}{\Delta P}$$
where Δ*C* is the change in capacitance of the sensor before and after compression, *C_1_* is the capacitance value after compression, *C_0_* is the initial capacitance value, Δ*P* is the applied compressive stress, *A* is the facing area of the electrodes, *d_1_* is the distance between the electrodes after compression, and *d_0_* is the distance between the electrodes at the initial stage, *ε_r2_* is the relative dielectric constant of the intermediate dielectric layer after compression, *ε_r1_* is the relative dielectric constant of the intermediate dielectric layer at the initial stage, and *ε_0_* is the vacuum dielectric constant.

The surface microstructure of the sensor can improve its sensitivity, because during the compression process, air is squeezed out from the microstructure, leading to an increase in the proportion of the middle dielectric layer in the overall middle layer. This results in an increase in the equivalent dielectric constant *ε_r_*_2_/*ε_r_*_1_ of the middle layer, which, in turn, increases the sensitivity of the sensor. Secondly, for basic surface microstructures such as circular columns, cones, pyramids, domes, square columns, and long micro-ridges, under the same compressive stress, the greater the compressive displacement, the smaller the distance between the electrodes, and the higher the sensitivity. Among them, cones, pyramids, and long micro-ridges have smaller contact areas during the compression process, leading to stress concentration at the tips, resulting in larger mechanical deformation, smaller electrode distances, and higher sensitivity. Furthermore, ladder-shaped microstructures, such as long micro-ridge ladders, cuboid ladder structures, and cylindrical ladder structures, have further improved compressibility. As a result, the sensitivity of these ladder structures is higher than that of the corresponding basic structures.

In summary, among the 9 surface microstructures, those with smaller contact areas at the top tend to have higher sensitivity than those with larger contact areas. Moreover, adding a ladder structure on the surface of a microstructure can further increase its sensing sensitivity. Among the long micro-ridge ladder, cuboid ladder, and cylindrical ladder structures, the cylindrical ladder structure is more isotropic, meaning that different axial orientations have minimal influence on the capacitance change, resulting in higher sensitivity. Based on this, this paper focuses on the cylindrical ladder structure and investigates the impact of each parameter of the structure on its sensitivity. In addition, we made a scanning electron microscope on these molds with microstructures manufactured by 3D printing to observe the surface morphology of the molds. In particular, scanning electron microscopy was performed on the cylindrical step structure prepared by the casting method that we are most concerned about. These can be found in Appendix A and Appendix A in the Appendix A.

## 3. Effect of Microstructure Intrinsic Dimensions on Sensing Sensitivity

When the sensor is subjected to external pressure, the plates typically do not change in area but move closer together, causing the distance between them to decrease. As a result, the dielectric layer is compressed, and air with a low dielectric constant is gradually expelled. The proportion of polymer to the total volume increases, leading to a further increase in its dielectric constant. After the microstructure on the surface has been created, air is introduced into the middle layer, which causes a decrease in the elastic modulus of the overall material. This results in the sensor being more sensitive to external forces and easier to deform.

Based on our previous experiments and simulations, we have identified the cylindrical ladder as the most effective microstructure design for a flexible capacitive pressure sensor. Further investigation was conducted on the various intrinsic dimensions of the cylindrical ladder, such as the diameter of the first step, the number of steps, the thickness of each step, the width of the steps, and the separation distance between the microstructure, as seen in Figure 4. Through this study, we were able to gain a deeper understanding of the influence of these intrinsic dimensions on the overall performance of the sensor.

The sensitivity of the sensing performance was studied by using the control variable method to investigate the effects of the 5 intrinsic dimensions of cylindrical ladder microstructures. Both simulation and experimental tests (seen in Figure 4b–f) have been carried out.

First of all, for the intrinsic dimension of the diameter of the first step, since the initial compression is mainly concentrated at the tip, the diameter of the first layer determines the contact area between the upper electrode and the microstructure interface during the early stages of compression. An increase in the diameter of the first step results in a larger contact area, lower compressibility, and reduced sensitivity. The diameter of the first step cannot be infinitely small since, in such cases, the electrode distance may become smaller than the designed distance due to bending of the first layer before the capacitive sensor undergoes compression.

Secondly, the thickness of each step together with the number of steps determines the total height of the sensor. Increasing the number of steps results in a higher total height of the microstructure, a larger distance between sensor electrodes, a decrease in initial capacitance, and an increase in sensitivity. On the other hand, increasing the total height of the microstructure decreases the equivalent elastic modulus of the intermediate dielectric layer, leading to higher compressibility, then to the sensitivity increase.

Thirdly, the separation distance between each microstructure determines the proportion of air in the microstructure layer. An increase in the separation distance results in a higher air volume ratio, which reduces the volume ratio of the dielectric material in the microstructure layer, ultimately increasing the compressibility of the intermediate dielectric layer and enhancing sensitivity. However, the separation distance cannot be too large, because the flexible electrodes in contact with the interface microstructures may collapse if the separation distance is too large.

Lastly, as the width of each step increases, the length of the bottom side gradually increases, so that the number of microstructures on the PDMS film of the same size decreases, the total contact area between the electrode and the microstructure decreases, the compressibility increases, and the sensitivity increases.

The insights gained from our investigation will be valuable in guiding the more detailed design of microstructures in future research. By exploring the intrinsic size of the microstructure, we can improve the performance of flexible capacitive pressure sensors and provide more accurate and reliable data.

## 4. Sensing Performance of Cylindrical Ladder Microstructure Capacitive Sensors

The presence of voids in the surface microstructure film helps to store and release energy in a reversible manner during the elastic deformation of the structural surface, reducing the influence caused by the viscoelastic behavior of the PDMS dielectric layer.

In the previous section, we analyzed the influence of the intrinsic dimensions of the cylindrical ladder microstructure on the sensitivity of the flexible capacitive pressure sensor. In this section, we explored the performance of the flexible capacitive pressure sensor with cylindrical ladder microstructure in more detail. The selection of the five intrinsic dimensions is based on the criterion of maximizing the sensitivity. Therefore, the selected parameters of the microstructure of the cylindrical steps of the dielectric layer in the sensor prototype are: the diameter of the first steps is 50 μm, the number of steps is 5, the thickness of each step is 50 μm, the width of the steps is 70 μm, and the separation distance between each microstructure is 1220 μm.

The response time of the flexible capacitive pressure sensor with the cylindrical ladder microstructure has been evaluated, as shown in Figure 5a. The response time refers to the time required for the sensor to respond to changes in pressure. The instron universal testing machine was used to apply an instantaneous displacement of 10 um to the sensor, and the loading and unloading speeds were all the way 500 μm/min. The response time is 21 ms when loading and 36 ms when unloading; the response time of the sensor was relatively short, which indicates that it has a fast response to pressure changes.

In order to verify its detection limit and resolution, pressures of 20 Pa, 40 Pa, and 60 Pa were, respectively, applied to the sensor. As shown in Figure 5b, when the above pressures are applied sequentially, the capacitance change rate shows a stepwise increase.

The hysteresis of the flexible capacitive pressure sensor with the cylindrical ladder microstructure has been evaluated. Hysteresis refers to the difference in sensor response when loading and unloading. The results in Figure 5c showed that the sensor had a low hysteresis of 12.567%, which indicates that it can quickly return to the original state after deformation.

To assess the stability of a sensor’s sensing performance under static load, the sensor is subjected to a force ranging from 10 to 50 kPa, which is maintained for 60 s, as illustrated in Figure 5d. The test results indicate that the capacitance change rate of the sensor remains stable under static loading at varying pressures. This suggests that the sensor can accurately detect and measure changes in pressure over time, without significant drift or loss of accuracy.

The response characteristics of the sensor under different pressure inputs have been analyzed. The sensor was subjected to 10, 30, and 50 kPa dynamic pressure inputs (10 loading and unloading cycles), and the corresponding changes in the capacitance were measured. The results in Figure 5e show that the output capacitance of the sensor increased with increasing pressure, which indicates that the sensor has reliable, stable, and repeatable sensing behavior.

Finally, we evaluated the durability of the sensor by subjecting it to repeated pressure cycles. The capacitive response to a compressive displacement of 300 μm was measured for 409 cycles. The results showed that the sensor had good durability, and its performance remained stable even after multiple pressure cycles, as shown in Figure 5f.

Besides, by sticking the flexible capacitive pressure sensor on molds with different bending radii for pressure measurement, we found that under different bending radii conditions (30 mm, 50 mm, 70 mm, and flat surface), the developed flexible capacitive pressure sensor has a stable output under different pressures, and the capacitance change rate of the output signal does not change due to the change of the bending radius, as shown in Appendix A, which proves that the flexible capacitive pressure sensor we developed has good pressure-capacitance sensing performance.

In conclusion, the flexible capacitive pressure sensor with the cylindrical ladder microstructure demonstrated excellent performance in terms of sensitivity, response time, hysteresis, durability, and stability.

## 5. Demonstration of the Microstructural Sensor

To accurately test rainfall, there are various methods and tools that can be utilized, such as rain gauges, weather stations, satellite images, and radar maps. However, each of these methods has its own limitations when it comes to real-time performance and accuracy. For example, rain gauges are specialized devices designed to measure the amount of rainwater collected over a specific period of time. They come in two common types: the funnel type and the tipping-bucket type. Although they can provide accurate measurements of rainfall, they require regular maintenance and can only measure precipitation in a limited area. The choice of method or tool for testing rainfall depends on the specific needs and limitations of the situation. Regardless of the method used, it is important to ensure accuracy and real-time performance to effectively monitor and predict rainfall.

Our flexible pressure sensor boasts exceptional sensitivity and operational stability, making it an ideal solution for monitoring rainfall frequency in real time. Additionally, the sensor can be conformally attached to outdoor leaves or building exterior walls, when sloped, allowing for precise and accurate measurement of rainfall. The flexibility of our pressure sensor allows it to conform to the contours of various surfaces, ensuring that it can be easily installed on different types of foliage and building exteriors. This versatility is crucial when it comes to monitoring rainfall in real time, as it ensures that the sensor can be easily positioned in the optimal location for measuring precipitation. With the ability to provide accurate, real-time data on rainfall frequency, our flexible pressure sensor has the potential to revolutionize the way we monitor and predict weather patterns. Whether it is for agricultural, environmental, or urban planning purposes, our sensor offers a reliable and cost-effective solution for rainfall monitoring.

To evaluate the sensing capability of our flexible pressure capacitive sensor in practical applications, we have attached the sensor to the surface of a plant leaf and used a liquid gun to simulate rainfall. The experiment was conducted with droplets falling at intervals of 15, 30, 45, and 60 s, as shown in Figure 6. The results of the experiment demonstrate that our flexible pressure capacitive sensor has a fast response time to the interval between droplets. The sensor capacitance quickly returns to its initial value as the droplet flows down the sensor surface, indicating the high sensitivity and stability of the sensor.

Overall, our flexible pressure capacitive sensor has the potential to revolutionize the way we monitor and predict weather patterns, providing a cost-effective and reliable solution for rainfall monitoring.

## Figures and Tables

**Figure 1 sensors-23-04323-f001:**
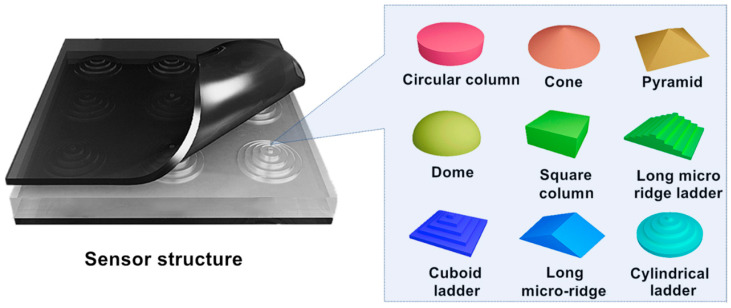
Nine types of microstructures studied in this paper.

**Figure 2 sensors-23-04323-f002:**
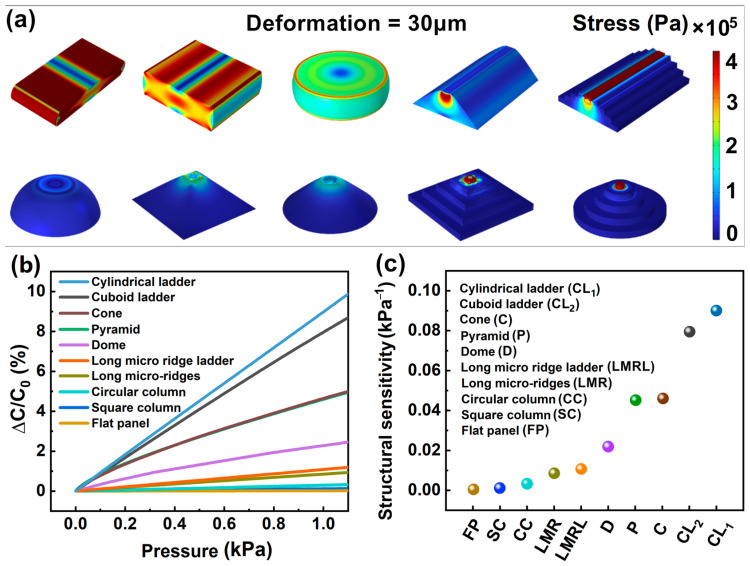
Simulation results of (**a**) Stress and stress distribution diagram of different microstructures under 30 μm strain; (**b**) Capacitance change rate of different microstructures under stress change; (**c**) Structural sensitivity of different microstructures.

**Figure 3 sensors-23-04323-f003:**
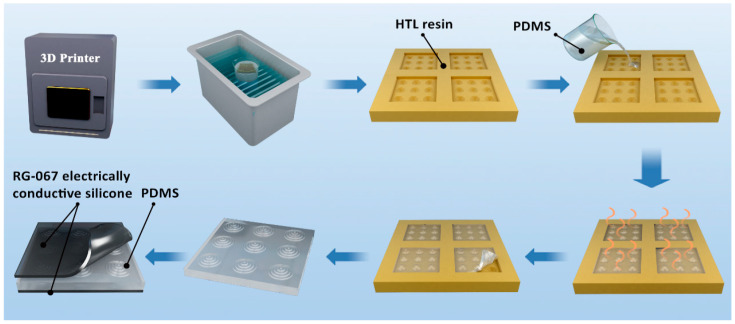
Fabrication process of micro-structural PDMS dielectric layer and flexible capacitive pressure sensor prototype.

**Figure 4 sensors-23-04323-f004:**
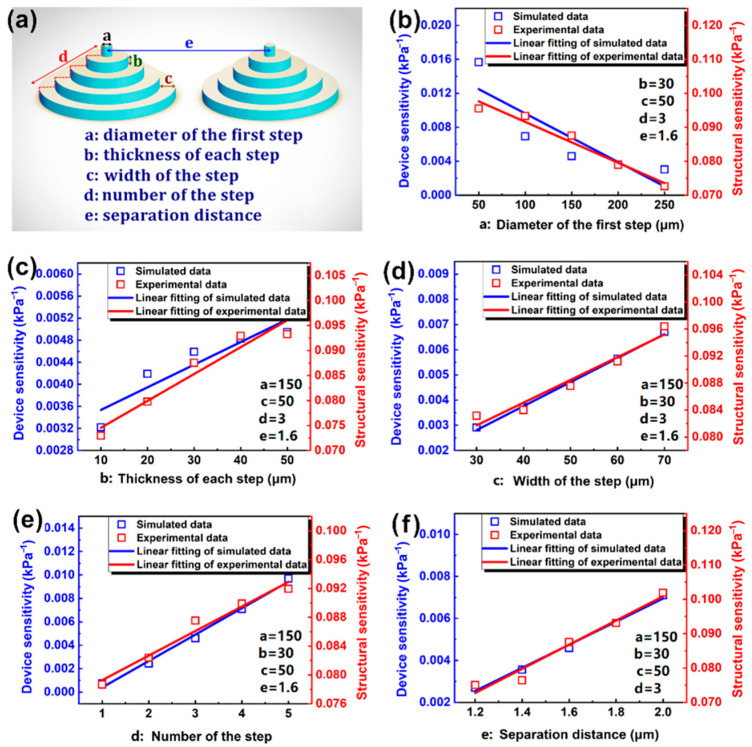
(**a**) The various intrinsic dimensions of the cylindrical ladder; The simulation structural sensitivity and experimental device sensitivity under the influence of (**b**) The diameter of the first steps; (**c**) The thickness of each step; (**d**) The width of the steps; (**e**) The number of steps; and (**f**) The separation distance between each microstructure.

**Figure 5 sensors-23-04323-f005:**
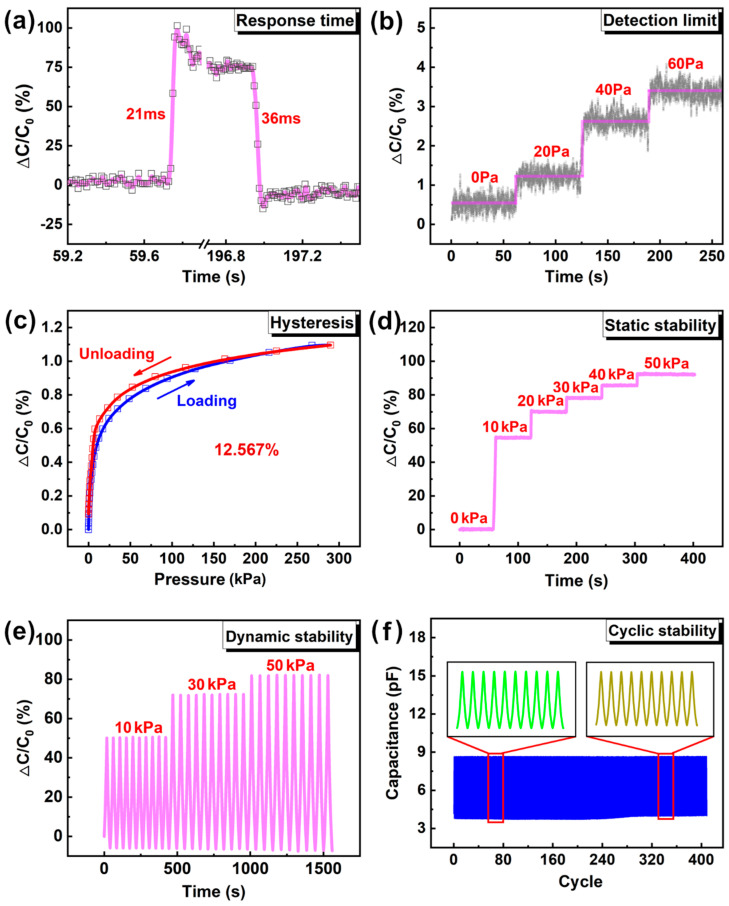
Sensing properties of pressure sensor. (**a**) Response time (21 ms) and recovery time (36 ms); (**b**) Detection limit of the sensor; (**c**) Hysteresis curves of loading/unloading process; (**d**) Capacitance response of sensors with different static loading; (**e**) Capacitance response of sensors with different dynamic loading; (**f**) Hysteresis curves of loading/unloading process for 409 cycles under compressive displacement of 300 μm.

**Figure 6 sensors-23-04323-f006:**
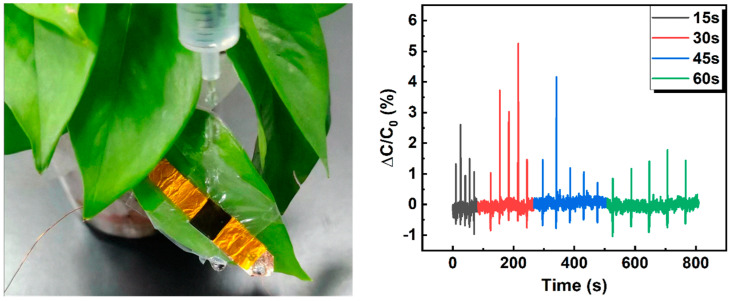
Demonstration of test rainfall by pressure sensor: (**left**) The actual picture of the sensor attached to the blade and applying water droplets; (**right**) Capacitance change rate with droplets falling at intervals of 15, 30, 45, and 60 s.

## Data Availability

Not applicable.

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
