# Peer review of "A Sensitivity-Optimized Flexible Capacitive Pressure Sensor with Cylindrical Ladder Microstructural Dielectric Layers"

_sensors, 2023, doi:10.3390/s23094323_

Round 1

Reviewer 1 Report

This manuscript has demonstrated a capacitive pressure sensor with cylindrical ladder microstructural dielectric layers that are fabricated with a 3D printer. This study could provide new ideas for the development of higher-performance, more reliable, and widely applicable flexible pressure sensors. However, some issues should be resolved before publishing.

1.        It is generally accepted as “kPa” rather than “KPa”.

2.        In Figure 2a, the unit N/m2 is suggested to transform into kPa to match the context.

3.        We know many factors will affect the stress distribution of a microstructure, such as size, volume, and so on. To facilitate a more accurate comparison, there should be one same parameter (e.g. volume) shared by these 9 microstructures. Otherwise, such comparisons are meaningless. 

4.        In Figure 4, a controlled variable method is used to compare the influence of five parameters of cylindrical ladder microstructures. only the changing parameters are presented for each plot, the values of each fixed parameter should also be presented in each figure (b to f).

5.        In Figure 5a, under what pressure conditions did the sensor show the response time and relaxation time of 21ms and 36 ms, respectively?

6.        In Figure 6, what is the limitation of droplet intervals?

Reviewer 2 Report

The authors reported the study of optimizing 3D-printed dielectric layers (with 9 microstructures) applied for flexible capacitive pressure sensors. The significance of this study is the comparison of COMSOL simulation data with experimental results to verify the best ladder structure of the dielectric layer. However, the manuscripts need some corrections, as listed below, before considering for publication.

1. There are many grammar errors. English correction is required.

2. Specify the dimensions (L W H) and spacings of microstructures (models) in Figs 1-2.

3. Looking at the sensor structure image in Fig 1, the microstructures appear on both sides of the dielectric layer. Meanwhile, the schematic fabrication process shows the microstructures formed on only one side of the dielectric layer. The authors must explain/correct that.

4. As the developed sensor is flexible, the authors need to show the strain response under various bending radii/angles. If possible, it is better if the authors can show the effect of compressive/tensile strain on the capacitive pressure sensing performance. 

Reviewer 3 Report

I think the manuscript is well organized and can be accepted for publication after some corrections are made.

- In Fig. 2(c), the structural sensitivity of the CL1 has a minimum value of almost zero and that of FP has the maximum value, which seems to be inconsistent with the description in the main text. Is the notation in Fig. 3(c) incorrect? If not, the definition of the ‘structure sensitivity’ needs to be clarified. The same is applied to Fig. S1. Although the word ‘the structure sensitivity’ is used in the text, the vertical axis in Figs. 1(c) and S1 is ‘Structural sensitivity’. Please match either one.

- The captions for Figures 4(c) and (d) are interchanged.

Round 2

Reviewer 1 Report

Manuscript is well revised.